# Modulation of Interlayer Nanochannels via the Moderate Heat Treatment of Graphene Oxide Membranes

**DOI:** 10.3390/polym16152200

**Published:** 2024-08-02

**Authors:** Na Meng, Xin Sun, Jinxin Liu, Jialing Mi, Xuan Chen, Rong Rong

**Affiliations:** Jiangsu Key Laboratory of Industrial Pollution Control and Resource Reuse, School of Environment Engineering, Xuzhou University of Technology, Xuzhou 221018, China; sunxin2708@163.com (X.S.); ljx2667495913@163.com (J.L.); 17751282269@163.com (J.M.); 1850519628@163.com (X.C.); rongrong114717@163.com (R.R.)

**Keywords:** GO membranes, d-spacing, moderate heat treatment, destaining and desalting

## Abstract

In response to the phenomenon of interlayer transport channel swelling caused by the hydration of oxygen-containing functional groups on the GO membrane surface, a moderate heat treatment method was employed to controllably reduce the graphene oxide (GO) membrane and prepare a reduced GO composite nanofiltration membrane (mixed cellulose membrane (MCE)/ethylenediamine (EDA)/reduced GO-X (RGO-X)). The associations of different heat treatment temperatures with the hydrophilicity, interlayer structure, permeability and dye/salt rejection properties of GO membranes were systematically explored. The results indicated that the oxygen-containing groups of the GO membrane were partially eliminated after heat treatment, and the hydrophilicity was weakened. This effectively weakened the hydration between the GO membrane and the water molecules and inhibited the swelling of the oxidized graphene membrane. In the dye desalination test, the MCE/EDA/RGO membrane exhibited an ultra-high rejection rate of over 97% for methylene blue (MB) dye molecules. In addition, heat treatment increased the structural defects of the GO membrane and promoted the fast passage of water molecules via the membrane. In pure water flux testing, the water flux of the membrane remained above 46.58 Lm^−2^h^−1^bar^−1^, while the salt rejection rate was relatively low.

## 1. Introduction

Water purification has turned into a global issue owing to industrial development, which has caused serious pollution in the environment. Membrane treatment is one of the most effective technologies for removing contaminants from water [1]. Nanofiltration membrane separation technology has received increasing attention because it can very effectively remove hardness, dissolved organics and heavy metals at a relatively low pressure, which is hard to achieve with traditional ultrafiltration [2,3,4].

Novel two-dimensional (2D) materials have opened up new avenues for innovation in nanofiltration membrane separation technology. Graphene oxide (GO), which is a classical 2D material, has gained considerable attention from researchers and shows potential as an assembly of high-performance membranes [5]. GO can be used to fabricate layered membranes through a simple and scalable layer-stacking technique [6]. It is very flexible in controlling layer spacing, charge and functionality, thus improving separation capacity and filtration efficiency. GO membranes have advantages in membrane separation, but prolonged immersion in water produces swelling, which is mainly related to some oxygen-containing groups on the GO surface [7]. To make GO membranes more useful for water separation, reduction [8], cross-linking [9], intercalation [10] and other strategies have been adopted to inhibit the swelling of GO membranes and enhance their stability.

Modulating the degree of reduction of GO membranes is conducive to improving their stability and salt rejection and shows the prospect of their use in nanofiltration [11]. Thermal reduction is a method commonly used to regulate the degree of reduction of GO membranes, which can effectively improve their stability and selective permeability. The hydrothermal reduction method used by Fan et al. [12], who synthesized reduced GO (RGO), regulated the degree of reduction of GO membranes by varying the hydrothermal reaction time. Compared to GO membranes, RGO membranes that were hydrothermally reduced exhibited superior permeation performance without sacrificing the rejection rate, which exceeded 99% for methylene blue (MB), crystal violet and Congo red. Zhang et al. [13] controlled the degree of reduction of GO by varying the temperature of the reaction during hydrothermal reduction. The GO membrane that was weakly reduced demonstrated a high water permeability of 56.3 Lm^−2^h^−1^bar^−1^, and the dye rejection increased to more than 95%. Yang et al. [14] prepared RGO membranes via vacuum filtration and regulated the degree of reduction of GO by changing the exposure time of GO membranes to hydriodic acid vapor for the purpose of controlling its nanochannel size. The results suggested that the nanochannel size decreased with an increasing degree of reduction of GO, and ion rejection was successfully tuned from 28.6% to 56.9%.

Designing two-dimensional (2D) lamellar nanochannels to allow more water to pass through and precisely controlling interlayer spacing to reject ions are necessary for GO membranes to have a high rejection and water permeability for desalination or water purification. Interlayer nanochannels are important factors affecting the water permeability and ion selectivity of layered membranes [15]. However, the heat treatment of GO membranes at ultra-high temperatures can disrupt the ordering of nanochannels in membranes and degrade membrane performance.

In order to effectively avoid the phenomenon of deterioration of membrane structure caused by the thermal reduction of graphene oxide membranes at ultra-high temperatures, in this paper, mild heat treatment was utilized to regulate the interlayer spacing of GO membranes. Changes in the structure and oxygen-containing functional groups of GO membranes after heat treatment were systematically characterized, with the expectation of optimizing membrane properties while reducing the damage to the interlayer nanochannels of membranes.

## 2. Materials and Methods

### 2.1. Materials

Monolayer GO powder was purchased from Jiangsu Pioneer Nanomaterials Technology Co., Ltd. (Nanjing, China). Sodium chloride (NaCl), magnesium sulfate (MgSO_4_) and sodium sulfate (Na_2_SO_4_) were provided by Sinopharm Chemical Reagent Co., Ltd. (Shanghai, China). Ethylenediamine (EDA) was bought from Shanghai Macklin Biochemical Technology Co., Ltd. (Shanghai, China). Mixed cellulose membrane (MCE, 0.22 μm) was obtained from Hangzhou Micropay Technology Co., Ltd. (Hangzhou, China). Methylene blue (MB) was provided by Shanghai Aladdin Biochemical Technology Co., Ltd. (Shanghai, China). The experimental water was pure water.

### 2.2. Preparation of MCE/EDA/RGO Composite Membranes

The preparation process of MCE/EDA/RGO-X membranes is shown in Figure 1. Specifically, 20 mg of GO powder was weighed, mixed with 500 mL of deionized water and then sonicated at 30 °C for 30 min to obtain 40 mg/L GO dispersion. Next, 40 mL of 40 mg/L GO dispersion and 100 mL of deionized water were mixed well, followed by the addition of 5 mL of EDA and a continuous 30 min ultrasonication at 30 °C to make an EDA/GO mixture. After that, the mixture was filtered via a vacuum filtration device onto an MCE by the vacuum filtration method to obtain a GO composite membrane. Finally, the prepared composite membranes were heat-treated via water bath heating at 25, 45 and 65 °C to prepare GO (MCE/EDA/RGO) composite membranes with different degrees of reduction. The composite membranes were labeled according to the temperature of heat treatment. For example, MCE/EDA/RGO-X indicated that the GO composite membrane was prepared by heat treatment at “X” °C.

### 2.3. Characterization of MCE/EDA/RGO Membranes

X-ray photoelectron spectroscopy (XPS) and Fourier transform infrared spectroscopy (FTIR) were used for analyzing the structures and functional groups of MCE/EDA/RGO membranes. The surface and cross-sectional morphology of the MCE/EDA/RGO membranes were observed using scanning electron microscopy (SEM) after spraying the membrane samples with gold (the thickness of the gold layer was between 10 and 20 nm). The crystal structure of GO nanosheets was analyzed by scanning the samples from 5° to 70° using an X-ray diffractometer (XRD). The d-spacing of the prepared GO membranes can be calculated using Bragg’s formula in Equation (1).
(1)$$d=\frac{n\lambda }{2sin\theta }$$
where *λ* represents the X-ray wavelength; *θ* stands for the diffraction angle; and *n* denotes the number of diffraction levels.

The hydrophilicity of the membranes was tested using the seated drop method with deionized water as the medium, and to reduce the chances of obtaining inaccurate experimental results, we performed at least two static contact angle tests for each sample. Atomic force microscopy (AFM) was used to observe the surface structure and roughness of the membranes. Changes in the internal defects of the GO composite membranes after heat treatment were analyzed using the Raman test.

### 2.4. Evaluation of Membrane Performance

Deionized water was used as the feed solution to measure the flux of MCE/EDA/RGO membranes and to assess their permeability. NaCl, Na_2_SO_4_ and MgSO_4_ at a concentration of 1000 ppm and a 10 ppm MB solution as the feed solution were utilized to test the rejection performance of the membranes.

Regarding the testing of flux and rejection performance, the flux of three samples of every membrane was tested under unchanged conditions, and then these three values were averaged. The membrane that had a water flux that was close to the average was selected for the rejection performance test. Membrane performance testing was performed using a dead-end filtration setup (Figure 2). The filtration system consists of a nitrogen pressurizer, a sealable water tank, a digital balance, a 300 mL filter cup, a magnetic stirrer and computerized online measurement software. The filter cup has an effective diameter of 0.032 m^2^ and a capacity of 300 mL. As for the measurement of rejection performance, the specific process of filtration was as follows: (1) First, a stable flux was reached by compacting the membrane at 2 bar for 2 h. (2) Then, the pressure was decreased to 1 bar, and the flux of purified water was recorded every 30 s. An average flux value was obtained by collecting at least 40 measurements. (3) Lastly, the salt and MB feed solution was used to replace the pure water. A stir bar was used for stirring the filter cup at 600 rpm to minimize concentration polarization, followed by the return of the pressure to 1 bar. In addition, 15 mL of filtrate was collected after each period of filtration.

Membrane permeability *J* (Lm^−2^h^−1^bar^−1^) was determined according to Equation (2). In the equation, *V* (L) represents the permeate volume per unit time; *A* (m^2^) stands for the effective filtration area of the membrane; *T* (h) denotes permeation time; and *P* (bar) refers to operating pressure.
(2)$$J=\frac{V}{T\times A\times P}$$

Measurements of the salt solutions were conducted using a conductivity meter to measure the conductivity of the stock solution and filtrate before filtration to evaluate the desalination performance of the membranes. The dye solution was gauged using an ultraviolet (UV)–visible spectrophotometer to evaluate the membrane rejection of the dye.

The rejection (*R*) of the salt ions and dye was calculated according to Equation (3), where *C*_0_ and *C*_1_ correspond to the concentrations of the feed solution and filtrate, respectively.
(3)$$R=\frac{C_{0}-C_{1}}{C_{0}}\times 100\%$$

## 3. Results and Discussion

### 3.1. FTIR Characterization of MCE/EDA/RGO Membranes

The MCE/EDA/RGO membranes were characterized using FTIR with increasing heat treatment temperature to examine the evolution law of the functional groups randomly distributed on the edge and basal surface of the GO membranes. As shown in Figure 3, the peak positions of the Gaussian peaks in the range of 1000–2000 cm^−1^ were 1056 cm^−1^ for alkoxy (C-O), 1272 cm^−1^ for epoxy (C-O-C), 1382 cm^−1^ (C-OH vibration) for carboxyl, 1625 cm^−1^ for carbon–carbon double bond (C=C) and 1735 cm^−1^ for carbonyl and carboxyl (C=O vibration). Hydroxyl stretching (-OH) was present in the broadband range of 3000–3500 cm^−1^. These characteristic peaks of GO are in agreement with previous reports in the literature [16]. As shown in Figure 3, the positions of the characteristic peaks corresponding to different wave numbers did not change, and oxygen-containing groups showed significant changes with the increase in the heat treatment temperature, with the absorption peaks of hydroxyl (-OH), epoxide (C-O-C) and alkoxyl (C-O) gradually diminishing. These changes indicated that GO membranes with different degrees of reduction can be obtained by changing the heat treatment temperature. In addition, a slight absorption peak can be seen near 1440 cm^−1^, which was due to the condensation of the amine group of EDA with the carboxyl group of GO and the simultaneous nucleophilic substitution reaction with the epoxy group to form C-N covalent bonds [17].

### 3.2. XPS Characterization of MCE/EDA/RGO Membranes

To understand the impact of changes in heat treatment temperature on the chemical properties of GO membranes, the elemental compositions of the MCE/EDA/RGO membranes at different temperatures were analyzed using XPS. As presented in Table 1, the O/C of MCE/EDA/RGO membranes gradually decreased from 0.41 to 0.34 with the rise in heat treatment temperature, which initially confirmed that the oxygen-containing functional groups on the surface of GO were gradually removed by reduction [18]. To deeply explore the mechanism by which heat treatment temperature influences MCE/EDA/RGO composite membranes, the simulated carbon (C1s) spectra of MCE/EDA/RGO membranes were decomposed. As illustrated in Figure 4, the MCE/EDA/RGO membrane can be decomposed into five peaks in the C1s region: C-C (284.6 eV), C-O/C-N (286.05 eV), C-O-C (286.75 eV), C=O (288.2 eV) and O-C=O (289 eV). The C-C bond (sp^2^ and sp^3^ domains) content increased with the increase in heat treatment temperature, but the strength of the C-O-C bonds (hydroxyl and epoxy groups) decreased significantly. Additionally, the N element gradually increased with the increase in heat treatment temperature according to a quantitative evaluation of its content in the composite membrane. This may be attributed to the fact that its formation was more favorable in the amidation reaction between GO and EDA at higher temperatures, which promoted the formation of C-N covalent bonds and the further removal of oxygen-containing functional groups.

### 3.3. Raman Testing of MCE/EDA/RGO Membranes

Carbon–carbon bonded states have high scattering efficiency, thus making Raman spectroscopy a powerful tool commonly used for characterizing carbon-based materials. In addition, Raman spectroscopy can be utilized to understand the structural changes in the GO membranes. Figure 5 shows a D peak near 1346 cm^−1^ and a G peak near 1600 cm^−1^, which represents the structure of oxidized or reduced GO. The D peak was ascribed to the respiration mode of the six-membered ring activated by structural defects, whereas the G peak was associated with the in-plane vibration of the sp^2^-hybridized carbon domains [19]. The ratio of the intensities of the D and G peaks (I_D_/I_G_) determined the structural defects (e.g., folds, wrinkles and nanoholes) and disorder in the GO nanosheets. As the heat treatment temperature increased, I_D_/I_G_ showed an increasing trend. This was attributed to the fact that MCE/EDA/RGO membranes were reduced and the oxygen-containing functional groups on GO membranes were gradually removed, which resulted in the tearing of the sp^2^ structure and the introduction of more structural defects into the membranes. The introduction of these defects was important for water molecule transport within the membranes.

### 3.4. SEM Characterization of MCE/EDA/RGO Membranes

The cross-section and surface of the MCE/EDA/RGO membranes were characterized using SEM. As shown in Figure 6, the MCE/EDA/RGO composite membranes at different heat treatment temperatures had a typical pleated surface [20]. The folds of the GO layer on the composite membrane surface decreased, and the membrane gradually became smooth with the increase in heat treatment temperature. However, the folds on the surface showed a tendency to grow gradually with a further increase in temperature to 65 °C, which may be related to the decrease in the number of functional groups on the GO membrane surface and the overlapping phenomenon of nanosheet layers. As shown in Figure 7, all MCE/EDA/RGO membranes exhibited a lamellar structure at the heat treatment temperatures of 25, 45 and 65 °C, which provided nanotransport channels for water molecule transport [21]. The fracture edges of the MCE/EDA/RGO-45 and MCE/EDA/RGO-65 membranes were looser relative to the MCE/EDA/RGO-25 membrane and disordered. Additionally, the thickness of the membranes decreased and then increased with the increase in temperature.

### 3.5. AFM Characterization of MCE/EDA/RGO Membranes

AFM was used to characterize the surface morphology and roughness (Ra) of the GO membranes at various heat treatment temperatures. The 3D surface AFM pictures of the GO membranes at various heat treatment temperatures are shown in Figure 8. As shown in Figure 9, the roughness of the MCE/EDA/RGO membrane decreased and then increased slightly with the elevation in heat treatment temperature. This illustrates that a proper elevation of temperature would still preserve the properties of GO membranes and would help fabricate a uniform membrane [19]. At higher temperatures, the GO membranes became regionally hydrophobic because additional oxygen-containing functional groups were removed from the surface. This made the GO prone to aggregation, which led to a decrease in free energy to form folds.

### 3.6. Water Contact Angle Testing of MCE/EDA/RGO Membranes

Membrane hydrophilicity was closely associated with permeation performance during water separation, which was assessed through the measurement of the contact angle on the GO membrane surface. The hydrophilicity of the GO membranes was closely related to the number of hydroxyl, carboxyl, and epoxy groups and other hydrophilic groups [22]. As shown in Figure 10, the MCE/EDA/RGO membranes had a water contact angle of 11.43°, 70.76° and 63.43° at the heat treatment temperatures of 25, 45 and 65 °C, respectively. The results indicated that the GO membranes were reduced and that these hydrophilic groups were partially removed with the rise in heat treatment temperature, which enhanced the hydrophobicity of the GO membranes. Notably, the MCE/EDA/RGO membranes showed a slight decrease in water contact angle at a heat treatment temperature of 65 °C. This may be attributable to the formation of more defects in the GO membranes at higher temperatures. The roughness of the membrane surface became larger, which provided more contact sites for water molecules [23]. This is in line with the characterization results of AFM.

### 3.7. XRD Characterization of MCE/EDA/RGO Membranes

An XRD analysis of the composite membranes was performed, and XRD diffractograms were obtained. The interlayer spacing of the MCE/EDA/RGO membranes in the dry state at different heat treatment temperatures was calculated using Bragg’s formula (see Equation (1)). The purpose was to probe the effect of heat treatment temperature on the interlayer structure of the MCE/EDA/RGO membranes. As shown in Figure 11a, the distribution and intensity of the corresponding GO peaks produced corresponding changes due to the different degrees of reduction of the GO membranes. At different heat treatment temperatures, the MCE/EDA/RGO membranes exhibited characteristic peaks at 8.162°, 10.269° and 9.985°, which corresponded to interlayer spacings of 1.082, 0.861 and 0.885 nm, respectively. As the heat treatment temperature increased, characteristic peaks were shifted to higher 2θ values, and the interlayer channels of the MCE/EDA/RGO membranes were narrowed; the channels became narrower overall. The variation in d-spacing of the membranes was further analyzed to more clearly present the micro-action of the heat treatment temperature on the interlayer structure of the membranes. As demonstrated in Figure 11b, the MCE/EDA/RGO membranes exhibited an increase in interlayer spacing from 0.861 to 0.885 nm with a slight expansion of the interlayer nanochannels when the heat treatment temperature rose from 45 to 65 °C. This may have occurred because the functional groups were removed and defects were formed on the GO substrate, which rendered the GO membranes loose and disordered.

### 3.8. Permeability of MCE/EDA/RGO Membranes

In the present study, a terminal filtration unit was used to investigate the impact of heat treatment temperature on the performance of the MCE/EDA/RGO membranes. As shown in Figure 12, the pure water permeability of the MCE/EDA/RGO membranes decreased from 104.07 to 46.58 Lm^−2^h^−1^bar^−1^ as the heat treatment temperature rose from 25 to 45 °C. This was caused by the narrowing of the interlayer nanochannels of the MCE/EDA/RGO membranes after the reduction of the membranes, which was in agreement with the XRD diffractogram (Figure 11) results of the MCE/EDA/RGO membranes. After heat treatment at 65 °C, the pure water permeability of the MCE/EDA/RGO membranes increased to 74.78 Lm^−2^h^−1^bar^−1^. The increase in water flux could have been engendered by an improvement in hydrophobic nonoxidizing sp^2^ structural domains, an increase in cracks and a reduction in the number of oxygen-containing groups following the slip flow theory [24,25].

### 3.9. Desalination Performance of MCE/EDA/RGO Membranes

The desalination performance of MCE/EDA/RGO membranes was tested using single salt solutions (Na_2_SO_4_, MgSO_4_ and NaCl). As shown in Figure 13, the MCE/EDA/RGO membranes rejected inorganic salts according to Na_2_SO_4_ > MgSO_4_ > NaCl, and the rejection of divalent anions was significantly higher than that of monovalent anions and divalent cations, which is in alignment with previous reports [26]. The separation mechanism of nanofiltration membranes was mainly linked to the resistance effect of spatial sites and electrostatic interactions [27]. In the ion transport process, high-valence co-ions are needed to overcome a larger interaction energy barrier relative to low-valence co-ions [28]. Graphene oxide membranes with a negative charge on the surface have a significant electrostatic repulsion effect on high-valence co-ions (SO_4_^2−^), which leads to an easy rejection of SO_4_^2−^. Meanwhile, Na^+^ was also rejected to maintain electroneutrality in the solution. Therefore, the MCE/EDA/RGO membranes had a higher rejection rate for Na_2_SO_4_. Since Mg^2+^ has a larger hydration radius than Na^+^ and Mg^2+^ transport was inhibited, MgSO_4_ had a higher rejection rate than NaCl. The MCE/EDA/RGO membranes showed an excellent desalination performance when the heat treatment temperature was 45 °C. This was attributed to the fact that the heat treatment reduced the GO membranes and narrowed the interlayer nanochannels. However, the deterioration of the MCE/EDA/RGO membrane structure was easily exacerbated by the heat treatment at higher temperatures, and the degree of membrane defects was increased, which decreased the ability of the membranes to intercept salt ions.

### 3.10. Decontamination Performance of MCE/EDA/RGO Membranes

To evaluate the dye removal effect of the MCE/EDA/RGO membranes at different heat treatment temperatures, the common MB dye was chosen for the rejection test. The dye removal effect of the nanofiltration membranes was related to the combined effect of adsorption, pore size sieving and the chargeability of the membrane surface [29,30,31]. As shown in Figure 14, the rejection of MB by the MCE/EDA/RGO membranes was 96.71%, 97.03% and 80.40% at temperatures of 25, 45 and 65 °C, respectively. This indicates that the structure of the MCE/EDA/RGO membranes can easily maintain its integrity and efficiently remove dye molecules at mild heat treatment temperatures (25 and 45 °C). As the heat treatment temperature was increased to 65 °C, the oxygen-containing functional groups on the surface of the graphene oxide membrane were gradually consumed, leading to the tearing of the sp^2^ structural domains, and the defects on the membrane surface were further enlarged, which helped the methylene blue molecules to rapidly permeate the membrane and thus exhibit a lower rejection rate. Therefore, graphene oxide composite membranes with excellent dye separation properties can be obtained by means of mild heat treatment, which has great potential for applications in the selective separation of dyes and salts in wastewater containing high levels of these compounds.

The molecular transport of the 2D material assembly selection layer occurred in both the vertical and horizontal directions (see Figure 15). The horizontal channel is composed of interlayer capillaries formed between adjacent nanosheets and occasional wrinkles generated by corrugated nanosheets. Vertical channels include gaps between the edges of the same layer of nanosheets, as well as inherent porous defects in the membrane. Interlayer channels play an important role in rapid selective molecular transport, and the transport process is influenced by the size, length and wall chemical properties of the interlayer channels [5,32]. During the heat treatment of the composite film using a water bath at different temperatures, the oxygen-containing functional groups at the edges of the oxidized graphene were reduced to varying degrees. During this process, local nanosheets without arrangements and several large voids were generated. This disordered laminar structure is illustrated in Figure 16. When the water bath temperature was above 45 °C, this disordered structure and the surface pores were more pronounced, which led to a higher flux and a lower dye retention effect for the MCE/EDA/RGO-65 membrane compared to the MCE/EDA/RGO-45 membrane.

## 4. Conclusions

In this study, the controlled reduction of GO nanosheets was demonstrated using mild heat treatment, and the effect of the degree of reduction on the structure and separation properties of GO membranes was investigated. The results showed that the mild heat treatment conditions favored the formation of better controlled interlayer channels and inhibited the solvation of GO membranes, and the GO interlayer channels overlapped at higher temperatures due to the removal of functional groups and the generation of in-plane defective holes. As the heat treatment temperature increased, the nanoscale modulation of the GO interlayer spacing from 1.082 to 0.861 nm occurred. In the nanofiltration performance test, the positively charged MB dye molecule with a molecular weight of 319.85 Da had an ultra-high rejection rate of more than 97%, while NaCl had a lower rejection rate of less than 10%, which will help promote the selective separation of dyes and salt ions by nanofiltration membranes in practical applications.

## Figures and Tables

**Figure 1 polymers-16-02200-f001:**
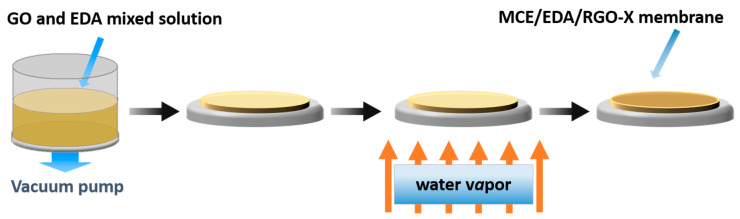
Schematic diagram of the preparation process of MCE/EDA/RGO-X membranes.

**Figure 2 polymers-16-02200-f002:**
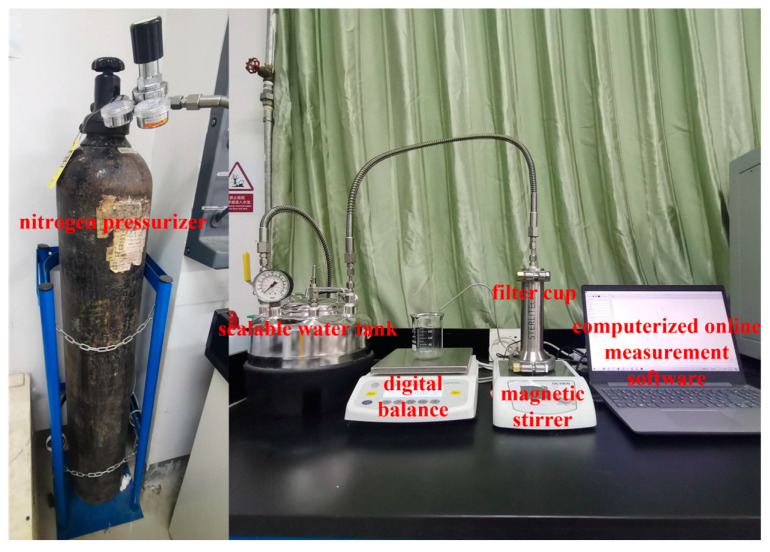
The dead-end filtration setup.

**Figure 3 polymers-16-02200-f003:**
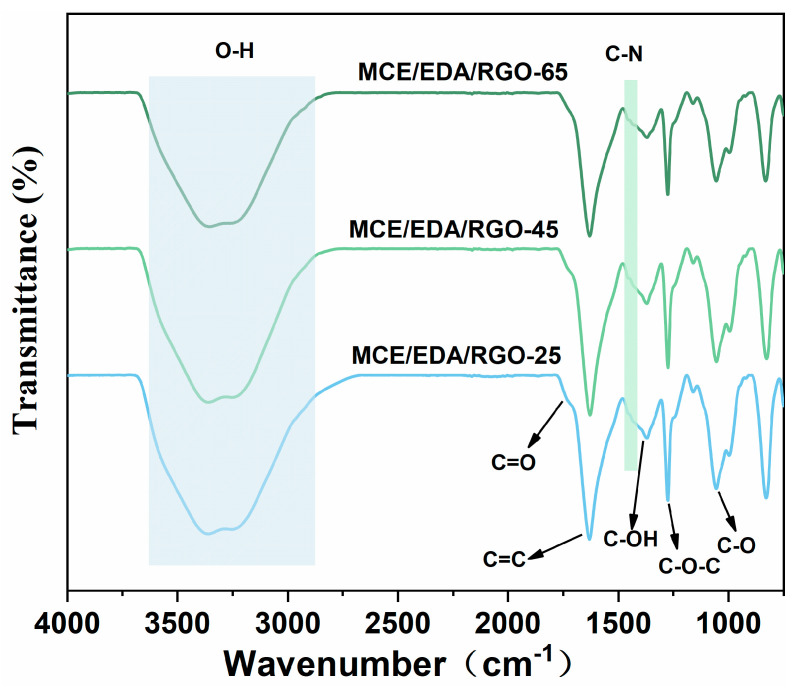
Infrared spectra of MCE/EDA/RGO at different heat treatment temperatures.

**Figure 4 polymers-16-02200-f004:**
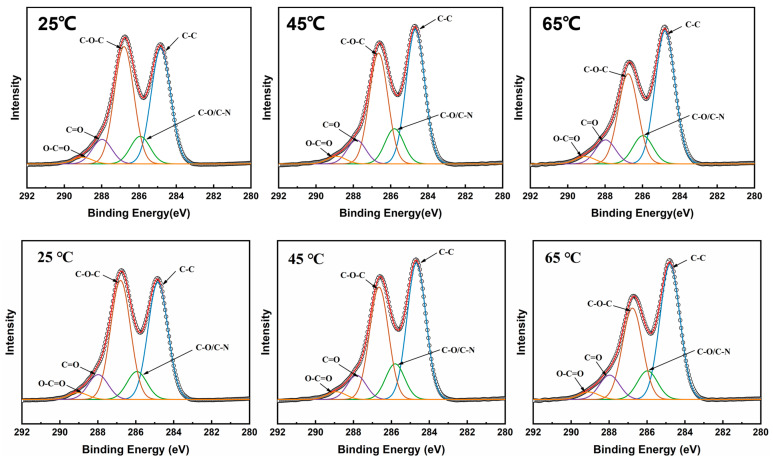
XPS elemental analysis of MCE/EDA/RGO membranes at different heat treatment temperatures in the region of C1s.

**Figure 5 polymers-16-02200-f005:**
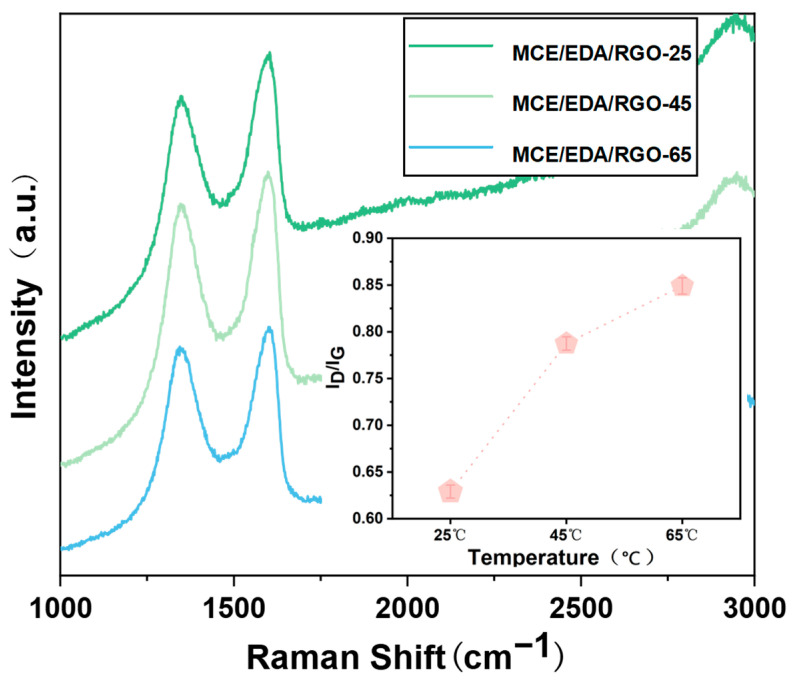
Raman spectra of MCE/EDA/RGO membranes at different heat treatment temperatures.

**Figure 6 polymers-16-02200-f006:**
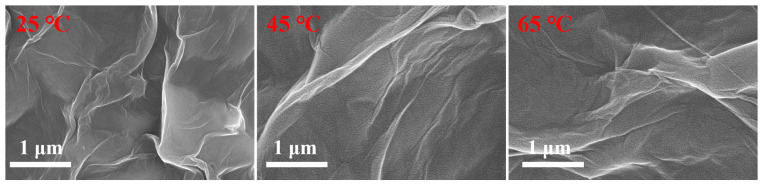
Surface characterization of MCE/EDA/RGO membranes at different heat treatment temperatures.

**Figure 7 polymers-16-02200-f007:**
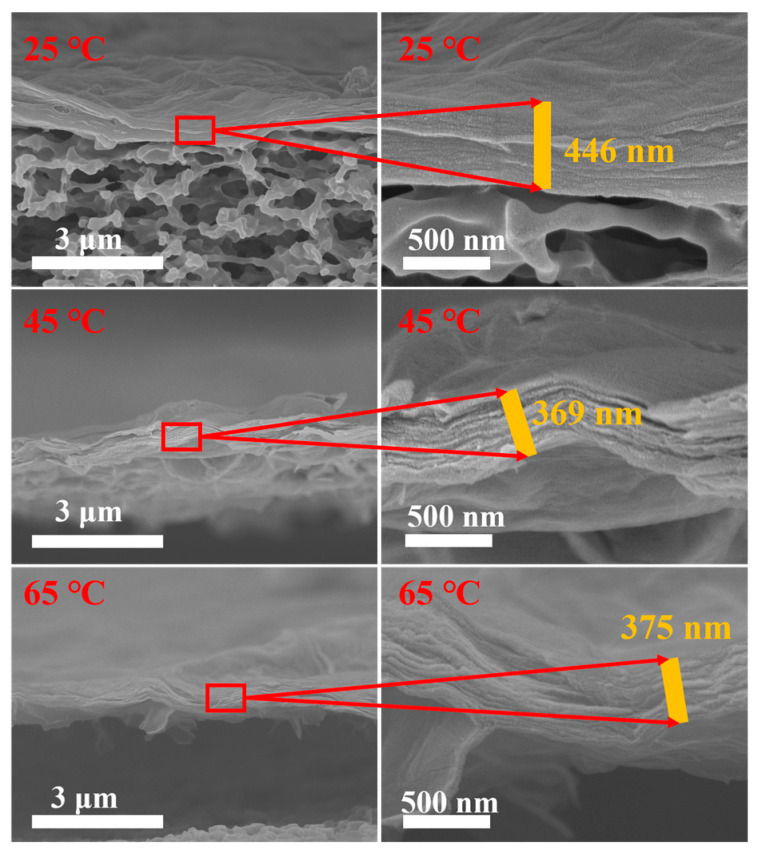
Cross-sectional characterization of MCE/EDA/RGO membranes at different heat treatment temperatures.

**Figure 8 polymers-16-02200-f008:**
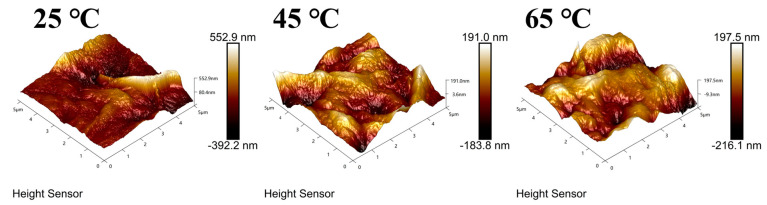
Surface 3D AFM pictures of MCE/EDA/RGO membranes at different heat treatment temperatures.

**Figure 9 polymers-16-02200-f009:**
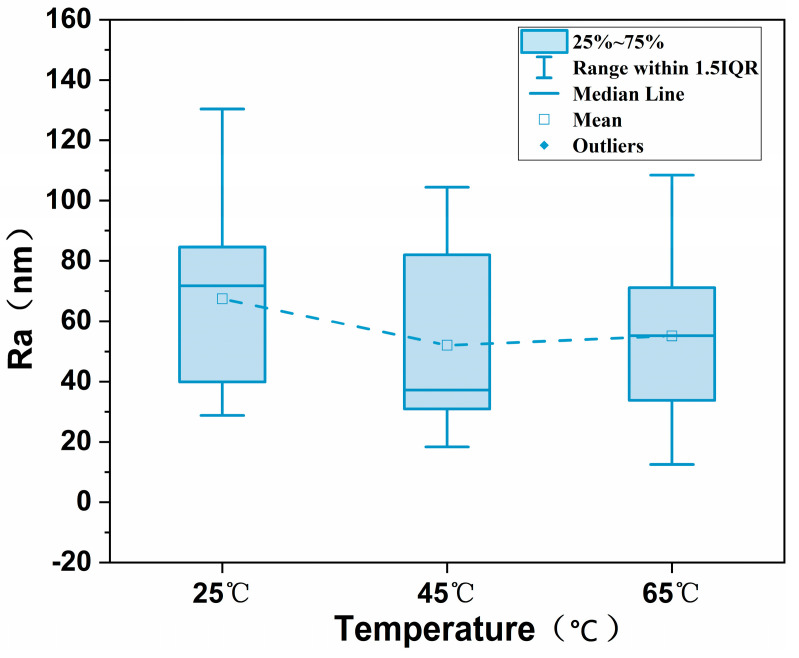
Roughness (Ra) of MCE/EDA/RGO membranes at different heat treatment temperatures.

**Figure 10 polymers-16-02200-f010:**
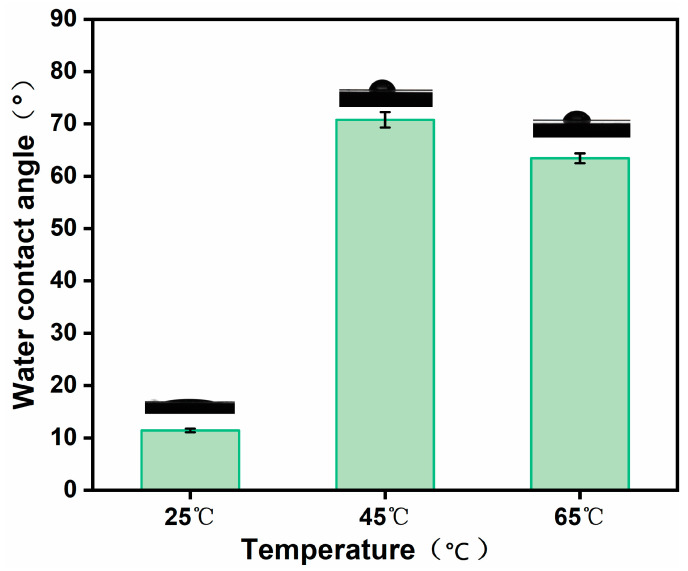
Water contact angles of MCE/EDA/RGO membranes at different heat treatment temperatures.

**Figure 11 polymers-16-02200-f011:**
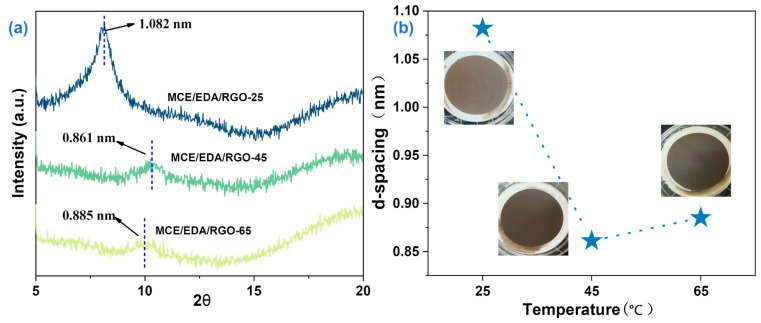
(**a**) XRD diffractograms of MCE/EDA/RGO membranes at different heat treatment temperatures. (**b**) Changes in the d-spacing of MCE/EDA/RGO membranes (The insert picture is the photo of the membrane).

**Figure 12 polymers-16-02200-f012:**
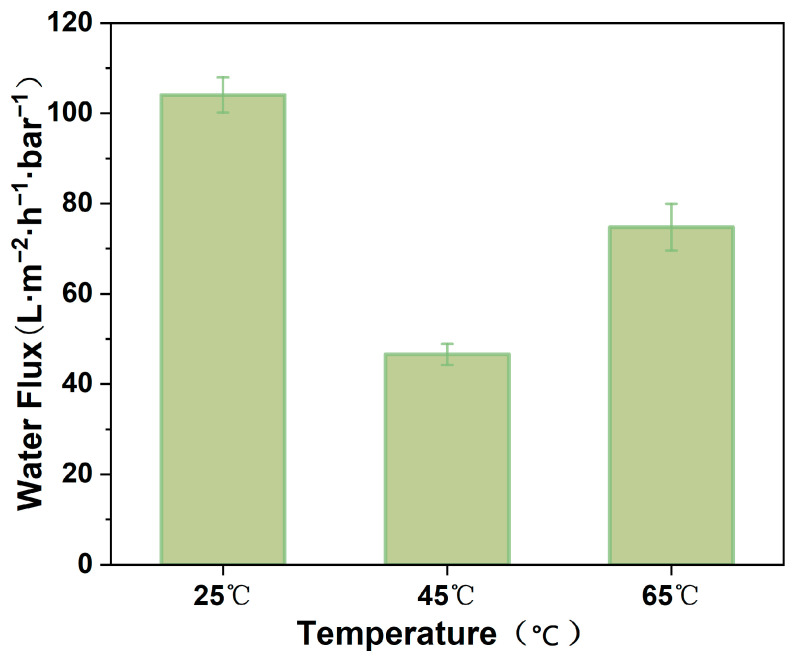
Pure water flux of MCE/EDA/RGO membranes at different heat treatment temperatures.

**Figure 13 polymers-16-02200-f013:**
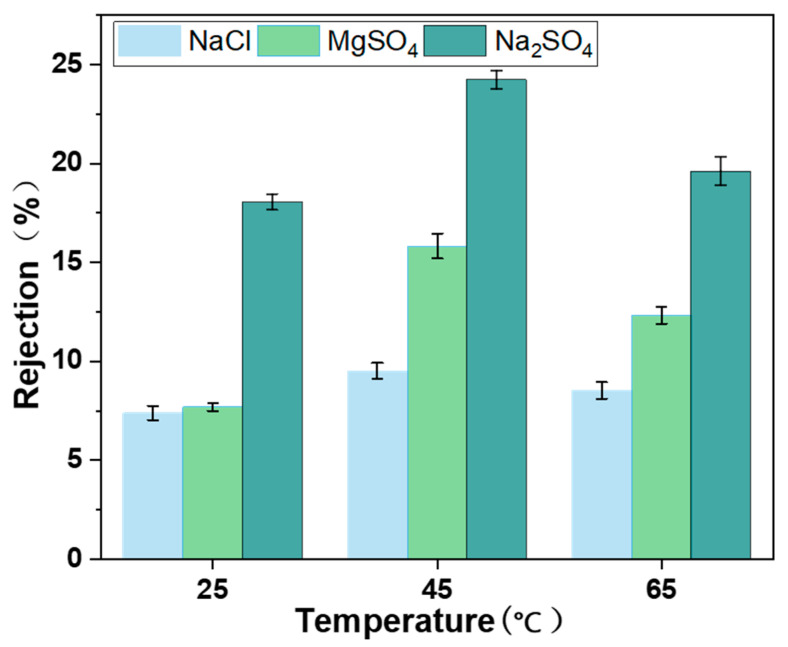
Salt ion rejection rates of MCE/EDA/RGO membranes at different heat treatment temperatures.

**Figure 14 polymers-16-02200-f014:**
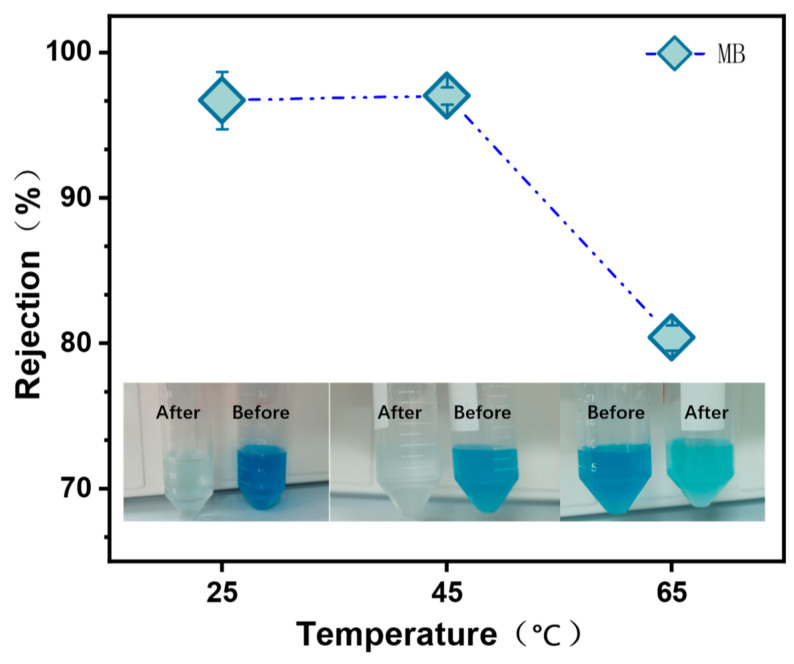
Rejection of MB by MCE/EDA/RGO membranes at different heat treatment temperatures.

**Figure 15 polymers-16-02200-f015:**
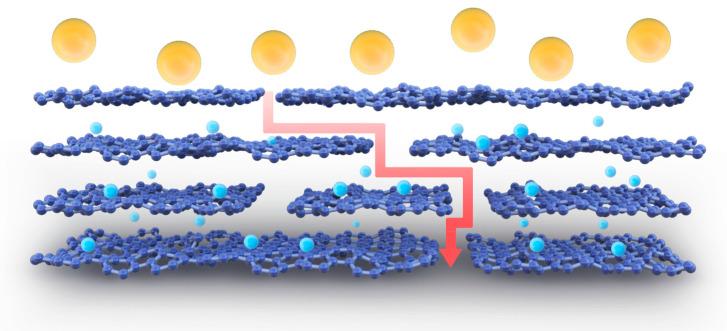
Schematic diagram of 2D material layered membrane transport (The yellow balls represent large ions and dyes, while the blue balls represent smaller).

**Figure 16 polymers-16-02200-f016:**
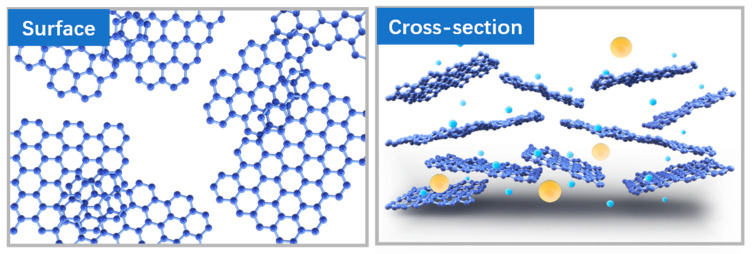
Surface defects and cross-sectional disordered structures in 2D channels (The yellow balls represent large ions and dyes, while the blue balls represent smaller).

**Table 1 polymers-16-02200-t001:** XPS elemental analysis of GO composite membranes at different heat treatment temperatures.

Samples	O%	N%	C%	O/C
MCE/EDA/RGO-25	27.71%	5.62%	66.67%	0.41
MCE/EDA/RGO-45	26.34%	5.81%	67.85%	0.38
MCE/EDA/RGO-65	23.7%	6.68%	69.62%	0.34

## Data Availability

The data presented in this study are available on request from the corresponding author. The data are not publicly available due to data are contained within the article.

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
