# Peer review of "Modulation of Interlayer Nanochannels via the Moderate Heat Treatment of Graphene Oxide Membranes"

_polymers, 2024, doi:10.3390/polym16152200_

Round 1

Reviewer 1 Report

Comments and Suggestions for Authors

The manuscript studies the effect of reduced graphene oxide (rGO)/ ethylenediamine composite on the filtration of methylene blue and some ionic salts. A membrane from an rGO composite with different reduction degrees has been fabricated, and its filtration performance is examined in detail. The study is focused on the reduction level of graphene oxide (GO), which is achieved by mild thermal treatment (up to 65 Celcius degrees) of the GO. The separation between the rGO layers in rGO reduces in higher when the treatment temperature increases. Also, based on XPS and FTIR results, treatment with higher temperatures reduces the functional groups. However, the membrane defect increases when treated at 65C, resulting in larger channels that are detrimental to the filtration performance. The manuscript systematically studies the membranes and is well-written. However, the following comments may improve it.

1-In page 11, the sentence "The negative charge on the GO membrane surface led to the easy rejection of GO membranes because of the 

obvious repulsive effect on high-valence co-ions." is unclear. The authors should rewrite it.

2-On page 11, section 3.10, the authors attribute the lower filtration capability of the GO composite reduced at 65C merely to its lower number of functional groups and disregard the membrane defects. If this is the case, the authors should explain why the membrane defects do not affect the prepared membrane's deficiency in absorbing MB, while it does for filtering ionic solutions. The reduction in the membrane treated at 65C is still considered mild, and based on FTIR and XPS data, many functional groups are still present in the membrane. 

3. The authors may analyse the membranes with a CHNS/O Elemental Analyzer after filtration of MB (if available). Analysing the amount of sulphur compound in the contaminated membranes may reflect the absorption capability of the rGO composites.

Author Response

We appreciate your professional comments on our article. As you may be concerned, there are several issues that need to be addressed. In response to your suggestions, we have made extensive corrections to our previous manuscript, which are listed below.

Comments 1:

In page 11, the sentence "The negative charge on the GO membrane surface led to the easy rejection of GO membranes because of the obvious repulsive effect on high-valence co-ions." is unclear. The authors should rewrite it.

Response 1:

Thank you for pointing this out. We agree with this comment. On page 11, section 3.9, we have revised the sentence as suggested.

In the ion transport process, high-valence co-ions are needed to overcome a larger interaction interaction energy barrier relative to low-valence co-ions [28].Graphene oxide membranes with a negative charge on the surface have a significant electrostatic repulsion effect on high valence co-ions (SO42-), which leads to the easy rejection of SO42-. Meanwhile, Na+ was also rejected to maintain solution electroneutrality. Therefore, MCE/EDA/RGO membranes had a higher rejection rate for Na2SO4. Since Mg2+ had a larger hydration radius than Na+, and Mg2+ transport was inhibited

Comments 2:

On page 11, section 3.10, the authors attribute the lower filtration capability of the GO composite reduced at 65C merely to its lower number of functional groups and disregard the membrane defects. If this is the case, the authors should explain why the membrane defects do not affect the prepared membrane's deficiency in absorbing MB, while it does for filtering ionic solutions. The reduction in the membrane treated at 65C is still considered mild, and based on FTIR and XPS data, many functional groups are still present in the membrane. 

Response 2:

Thank you for pointing this out. We agree that the reduced rejection of methylene blue molecules by GO membranes at 65°C is also related to membrane defects. As suggested, we have modified the sentence on page 12, section 3.10, to highlight the importance of defects on membrane performance. In addition, in the study of Huang et al [1], we learned that the formation of GO membrane defects can be attributed to the removal of oxygen-containing functional groups from the membrane surface. Further deoxygenation of graphene oxide membranes at 65 °C leads to the tearing of the sp2 structural domains of the graphene oxide structure, some carbon molecules are removed from the graphite structure, and the defects of the membranes are further enlarged, a conclusion that can be found in the Raman test section of this paper on page 6, section 3.3. Although FTIR and XPS tests showed that more functional groups still existed on the surface of the heat-treated GO membranes, further quantitative analysis of the XPS elemental composition showed that the percentage of oxygen decreased with the increase of the heat-treatment temperature, which confirms that the removal of oxygen-containing functional groups is closely related to the formation of membrane defects.

[1] H.-H. Huang, K.K.H. De Silva, G. Kumara, M. Yoshimura, Structural evolution of hydrothermally derived reduced graphene oxide, Scientific reports 8(1) (2018) 6849.

Fig. 14, the rejection of MB by MCE/EDA/RGO membranes was 96.71%, 97.03% and 80.40% at a temperature of 25, 45 and 65 °C, respectively. This indicates that the structure of MCE/EDA/RGO membranes can easily maintain its integrity and be efficiently removed for dye molecules at mild heat treatment temperatures (25 and 45 °C). As the heat treatment temperature was increased to 65 °C, the oxygen functional groups on the surface of the graphene oxide membrane were gradually consumed leading to the tearing of the sp2 structural domains, and the defects on the membrane surface were further enlarged, which helped the methylene blue molecules to rapidly permeate through the membrane and thus exhibit a lower rejection rate. Therefore, graphene oxide composite membranes with excellent dye separation properties can be obtained by means of mild heat treatment, which has great potential for application in selective separation of dyes and salts in highly saline dye wastewater.

Comments 3:

The authors may analyse the membranes with a CHNS/O Elemental Analyzer after filtration of MB (if available). Analysing the amount of sulphur compound in the contaminated membranes may reflect the absorption capability of the rGO composites.

Response 3:

Thank you for pointing this out. We agree with you that the content of sulfur compounds in the contaminated membranes analyzed by CHNS/O Elemental Analyzer during MB filtration reflects the absorptive capacity of the rGO composites. We will use the CHNS/O Elemental Analyzer to analyze the membranes in the next study to ensure the integrity of the study.

Reviewer 2 Report

Comments and Suggestions for Authors

The manuscript ‘Modulation of interlayer nano-channels via the moderate heat treatment of graphene oxide membranes’ refers to a very interesting issue involving the prepare of a new type of membranes, but it is necessary to preparing the revised version of this article and I have a few comments that the authors should consider in preparing the revised version.

Necessary, the authors should be added the aim of this paper in ‘Introduction’ section.

In the Materials and Methods section should be explained:

- Why these process parameters and composition are used during membrane preparation?

- Why this composition of model solution was used in experiments?

- How thickness of the gold layer the samples were sprayed prior to the SEM observation?

- What solutions were used in determining the contact angle?

- Authors should be add the experimental set-up in the ‘Materials and Methods’ section.

In the Results and Disccusion section should be supplemented:

- How is potential application of this type of membranes?

- The stability of working is one of the most important factor to consider during preparation of new type of membranes. Please, explain what the stability of graphene oxide membrane developed in this work during filtration.

Author Response

We appreciate your professional comments on our article. As you may be concerned, there are several issues that need to be addressed. In response to your suggestions, we have made extensive corrections to our previous manuscript, which are listed below.

Comments 1:

Necessary, the authors should be added the aim of this paper in ‘Introduction’ section.

Response 1:

Thank you for pointing this out. We agree with this comment. In introduction (page 2, line 66), we added the research objectives of this paper as suggested.

In order to effectively avoid the phenomenon of deterioration of membrane structure caused by thermal reduction of graphene oxide membranes at ultra-high temperatures, in this paper, mild heat treatment was utilized to regulate the interlayer spacing of GO membranes. Changes in the structure and oxygen-containing functional groups of GO membranes after heat treatment were systematically characterized, with the expectation of optimizing membrane properties while reducing the damage to the interlayer nano-channels of membranes.

Comments 2:

In the Materials and Methods section should be explained:

- Why these process parameters and composition are used during membrane preparation?

Response 2:

Thank you for pointing this out. In the pre-preparation stage of membrane preparation, we found that graphene oxide powder was not fully dispersed in water, so we used ultrasound to ensure that graphene oxide powder was fully dispersed in water. For the selection of graphene oxide membrane concentration, higher concentration of graphene oxide solution is not favorable for the transport of water molecules inside the membrane, while lower concentration of graphene oxide membrane shows very low ionic interception, which is the parameter selected through our pre-preparation exploration. By adding 5 ml of ethylenediamine as a cross-linking agent can cross-link with graphene oxide is beneficial for the membrane to maintain its integrity during the test. In addition, in order to minimize the damage to the membrane structure by thermal reduction, we established 25, 45, and 65 °C to explore the precise construction of the membrane structure at mild temperatures.

Comments 3:

In the Materials and Methods section should be explained:

- Why this composition of model solution was used in experiments?

Response 3:

Thank you for pointing this out. In order to explore the effect of temperature on membrane performance, we choose three salt and one dye molecules to test the rejection performance of the membrane for salt and dye after heat treatment. Here we used a solution of a single component salt and a solution of a single component dye, and the selective permeability of the membrane to the salt/dye can be reflected by testing the rejection rate of the membrane to different salts and dyes.

Comments 4:

In the Materials and Methods section should be explained:

- How thickness of the gold layer the samples were sprayed prior to the SEM observation?

Response 4:

Thank you for pointing this out. On page 3, section 2.3 we added the thickness of the gold layer.

X-ray photoelectron spectroscopy (XPS) and Fourier transform infrared spectroscopy (FTIR) were used for analyzing the structures and functional groups of MCE/EDA/RGO membranes. The surface and cross-sectional morphology of the MCE/EDA/RGO membranes were observed by scanning electron microscopy (SEM) after spraying the membrane samples with gold (the thickness of the gold layer was between 10-20 nm). The crystal structure of GO nanosheets was analyzed by scanning the samples from 5° to 70° by an X-ray diffractometer (XRD).

Comments 5:

In the Materials and Methods section should be explained:

- What solutions were used in determining the contact angle?

Response 5:

Thank you for pointing this out. We used deionized water as the medium for the contact angle test. On page 3, section 2.3 we provide additional instructions for the contact angle test.

The hydrophilicity of the membranes was tested by the seated drop method using deionized water as the medium, and to avoid the chance of the experimental results, we performed at least two static contact angle tests for each sample. Atomic force microscopy (AFM) was used to observe the surface structure and roughness of membranes. Changes in the internal defects of GO composite membranes after heat treatment were analyzed by the Raman test.

Comments 6:

In the Materials and Methods section should be explained:

- Authors should be add the experimental set-up in the ‘Materials and Methods’ section.

Response 6:

Thank you for pointing this out. We agree with this comment. On page 11, section 2.4, we have added a diagram of the experimental setup to help the reader better understand it. As shown in the figure below.

Figure 2. The dead-end filtration set-up.

Comments 7:

In the Results and Disccusion section should be supplemented:

- How is potential application of this type of membranes?

Response 7:

Thank you for pointing this out. We agree with this comment. We added the potential for membrane applications on page 12, section 3.10.

for dye molecules at mild heat treatment temperatures (25 and 45 °C). As the heat treatment temperature was increased to 65 °C, the oxygen functional groups on the surface of the graphene oxide membrane were gradually consumed leading to the tearing of the sp2 structural domains, and the defects on the membrane surface were further enlarged, which helped the methylene blue molecules to rapidly permeate through the membrane and thus exhibit a lower rejection rate. Therefore, graphene oxide composite membranes with excellent dye separation properties can be obtained by means of mild heat treatment, which has great potential for application in selective separation of dyes and salts in highly saline dye wastewater.

Comments 8:

- The stability of working is one of the most important factors to consider during preparation of new type of membranes. Please, explain what the stability of graphene oxide membrane developed in this work during filtration.

Response 8:

Thank you for pointing this out. We agree with this comment. In this work, we reduced the oxygen-containing functional groups on the surface of graphene oxide by means of mild heat treatment, which inhibited the water molecules from combining with the oxygen functional groups, alleviated the swelling phenomenon of graphene oxide film in contact with water, and avoided the collapse of the lamellar structure of graphene oxide film to some extent. In addition, the use of ethylenediamine as a cross-linking agent can have a nucleophilic substitution reaction with the graphene oxide membrane, forming C-N covalent bonds between adjacent graphene oxide lamellae, interlocking the graphene oxide layers with each other, and further constructing a stable interlayer nanochannel. These methods greatly improved the stability of graphene oxide membranes during filtration. In the next study, we will focus on the stability of the membrane.

Round 2

Reviewer 1 Report

Comments and Suggestions for Authors

Thank you for considering my comments and reflecting them in the revised manuscript.